# Perioperative Therapy for Non-Small Cell Lung Cancer with Immune Checkpoint Inhibitors

**DOI:** 10.3390/cancers13164035

**Published:** 2021-08-10

**Authors:** Junichi Soh, Akira Hamada, Toshio Fujino, Tetsuya Mitsudomi

**Affiliations:** Division of Thoracic Surgery, Department of Surgery, Kindai University Faculty of Medicine, 377-2 Ohno-higashi, Osaka-Sayama, Osaka 589-8511, Japan; a-hamada@med.kindai.ac.jp (A.H.); t-fujino@surg.med.kindai.ac.jp (T.F.); mitsudom@med.kindai.ac.jp (T.M.)

**Keywords:** lung cancer, immune checkpoint inhibitor, perioperative therapy, neoadjuvant therapy, adjuvant therapy

## Abstract

**Simple Summary:**

This review presents the development of perioperative treatment using immune checkpoint inhibitors (ICIs) in patients with resectable non-small cell lung cancers. There are several ongoing phase 3 trials for adjuvant and neoadjuvant ICI therapies. The results of the adjuvant (IMpower010 trial) and the neoadjuvant (Checkmate 816 trial) ICI phase 3 trials have shown prolonged disease-free survival and increased pathological complete response rate, respectively. Based on the hypothesis that ‘preoperative ICI treatment, especially in combination with conventional chemotherapy, promotes a higher immune response because of preservation of the immune environment’, neoadjuvant trials using ICIs and conventional chemotherapy in combination are currently being conducted more frequently than adjuvant ICI trials. Multimodality approaches using chemoradiotherapy and new ICI agents are also being examined in several phase 2 trials. To maximise ICI therapy’s efficacy and to minimise futile administration, methodologies for predicting and monitoring the therapeutic effects, such as detecting minimal residual disease, need to be established.

**Abstract:**

The emergence of immune checkpoint inhibitors (ICIs) has dramatically changed the treatment landscape for patients with metastatic non-small cell lung cancer (NSCLC). These achievements inspired investigators and pharmaceutical companies to conduct clinical trials in patients with early-stage NSCLC because both adjuvant and neoadjuvant platinum-based doublet chemotherapies (PT-DCs) showed only a 5% improvement in 5-year overall survival. IMpower010, a phase 3 trial (P3), showed that adjuvant PT-DC followed by maintenance atezolitumab significantly prolonged disease-free survival over adjuvant PT-DC alone (hazard ratio, 0.79; stage II to IIIA). Since conventional therapies, including chemotherapy and radiotherapy, can promote immunogenic cell death, releasing tumour antigens from dead tumour cells, ICI combination therapies with conventional therapies are widely proposed. The Checkmate 816 trial (P3) indicated a significantly higher pathological complete response rate of neoadjuvant nivolumab/PT-DC combination therapy than of neoadjuvant PT-DC alone (odds ratio, 13.9, for stage IB to IIIA). Detection of circulating tumour DNA is highly anticipated for the evaluation of minimal residual disease. Multimodal approaches and new ICI agents are being attempted to improve the efficacy of ICI treatment in phase 2 trials. This review presents the development of perioperative treatment using ICIs in patients with NSCLC while discussing problems and perspectives.

## 1. Introduction

Lung cancer remains the leading cause of cancer-related mortality worldwide [1]. Only a small fraction of patients can be treated with curative intent. Although surgery offers the best chance for a cure, the 5-year overall survival (OS) rates of patients who have to undergo pulmonary resection are unsatisfactory, with rates of 68%, 60%, 53%, and 36% for pathological stages IB/IIA/IIB/IIIA, respectively [2].

Cancer recurrence risk reduction, especially for distant recurrence, is essential for patients to achieve long-term survival after surgery. Distant recurrence occurs due to the progression of minimal residual disease (MRD), which is considered to be metastasised cancer cells that are undetectable in imaging studies prior to surgery. The current standard treatment modality for patients with pathological stage II to III non-small cell lung cancer (NSCLC) is adjuvant therapy using platinum-based doublet chemotherapy (PT-DC). However, the lung adjuvant cisplatin evaluation (LACE) meta-analysis of five randomised phase 3 trials reported that adjuvant PT-DC could improve 5-year survival by only 5.4% in patients after complete resection of NSCLC [3].

The advent of immune checkpoint inhibitors (ICIs) targeting programmed cell death 1 (PD-1), programmed death ligand 1 (PD-L1), and cytotoxic T-lymphocyte-associated protein 4 (CTLA4) has led to a durable response and improved prognosis in patients with metastatic lung cancer [4,5,6]. In addition, the PACIFIC trial reported that concurrent chemoradiation followed by maintenance therapy with durvalumab, a PD-L1 antibody, prolonged progression-free survival (PFS) and OS. Hence, this regimen has become the standard of care for patients with unresectable stage III NSCLC [7,8]. Thus, the application of ICIs to patients with early-stage lung cancer has been actively pursued. This article summarises perioperative treatments using ICIs in patients with NSCLC and discusses future perspectives.

## 2. Current Status of Perioperative Therapy

The evidence for adjuvant chemotherapy using cytotoxic agents has been well established. In 2008, the LACE meta-analysis of five phase 3 trials [3] showed that adjuvant chemotherapy using PT-DC significantly improved OS in completely resected patients with NSCLC (hazard ratio (HR) = 0.89, 95% confidence interval (CI): 0.82–0.96, *p* = 0.005). Furthermore, a subset analysis showed that adjuvant therapy using PT-DC improved OS in stage II (HR = 0.83; 95% CI: 0.73–0.95) and stage III (HR = 0.83, 95% CI: 0.72–0.94), but it was determined to be harmful in stage I NSCLC.

The evidence for the neoadjuvant chemotherapy results, although insufficient compared to that for adjuvant therapy, through a review and meta-analysis conducted by the NSCLC Meta-analysis Collaborate Group, showed that neoadjuvant chemotherapy followed by surgery for stage I to III NSCLC improved the 5-year OS by 5% (40–45%) (HR = 0.87, 95% CI: 0.78–0.96, *p* = 0.007) compared to surgery alone [9].

Limited evidence is available regarding the efficacy of the induction CRT followed by surgery. However, the INT0139 study, a phase 3 trial, compared the standard of care radical chemoradiotherapy (CRT) with induction CRT (45 Gy) followed by surgery for pathologically proven cN2 resectable NSCLC [10]. The trial reported that in an exploratory subset analysis, pneumonectomy after CRT induction was associated with a 26% treatment-related mortality rate and a worse OS than radical CRT. In contrast, lobectomy after CRT induction was associated with a 1% treatment-related mortality rate and significantly better OS than radical CRT (median OS, 33.6 vs. 21.7%, *p* = 0.002).

## 3. Initiation of Clinical Trials for Perioperative Therapy by ICI

ICIs improve the prognosis of patients with metastatic NSCLC [4,5,6,11,12,13]. These achievements have encouraged the introduction of immunotherapy as an adjuvant, neoadjuvant, or both for patients with earlier-stage lung cancer.

Several large-scale phase 3 studies are underway in an adjuvant setting, investigating the efficacy of ICIs after complete resection in patients with pathological stage IB to IIIA NSCLC (Table 1). There are currently at least seven phase 3 studies, including five ICI monotherapies and two combination therapies for ICI and conventional chemotherapy, estimated to accrue 5347 patients, in progress.

Recently, the main findings of the IMpower010 trial revealed that adjuvant chemotherapy followed by maintenance with atezolizumab showed the significant prolongation of disease-free survival (DFS) in patients with PD-L1 TC ≥1% (per SP 263) stage II to IIIA (HR, 0.66; 95% CI, 0.50–0.88) NSCLC [14]. Hierarchical analysis showed a significant prolongation of DFS in patients with all-randomised stage II to IIIA (HR, 0.79; 95% CI, 0.64–0.96) and in the intention-to-treat population with stage IB to IIIA (HR, 0.81; 95% CI, 0.67–0.99). The 3-year DFS rate was 55.7% in patients with all-randomised stage II to IIIA who received maintenance atezolizumab compared to the 49.4% in those who did not receive maintenance atezolizumab (*p* = 0.02). Among patients with all-randomised stage II to IIIA, under atezolizumab administration, patients with high PD-L1 expression of TC ≥ 50% (HR 0.43; 95% CI, 0.27–0.68) received the greatest benefit in terms of DFS. No survival benefit was observed in patients with the PD-L1 expression of TC < 1% (HR, 0.97; 95% CI, 0.72–1.31) after maintenance atezolizumab administration.

## 4. Neoadjuvant vs. Adjuvant

There is a possibility that neoadjuvant therapy may control micrometastases in the early phases and may offer an opportunity to evaluate drug sensitivity. Thus, it could be used as a guide in determining the postoperative regimen (Figure 1). Adjuvant therapy may or may not be performed under a reduced drug regimen if the patients’ performance status worsens after surgery. Neoadjuvant therapy can be performed with good compliance. However, neoadjuvant therapy may cause increased postoperative complications and treatment-related adverse events (TRAE), leading to delays in surgery or inoperability [15,16].

T cells are activated by the recognition of the presented tumour antigen. They travel through the lymphatic stream and the bloodstream to the primary and metastatic sites and exert anti-tumour effects. Hence, it has been argued that neoadjuvant ICI therapy may be more effective than adjuvant ICI therapy because lymphatic and blood flow between the tumour and regional lymph nodes are maintained in neoadjuvant therapy but not in adjuvant therapy [17]. These hypotheses were experimentally examined by comparing neoadjuvant and adjuvant ICI therapies using a mouse subcutaneous tumour transplantation model [18]. Mice treated with neoadjuvant ICI therapy had longer survival than those treated with adjuvant ICI therapy did.

In clinical trials for neoadjuvant and adjuvant therapies, the gold standard for the primary endpoint is OS. Since it takes a long time to obtain the final OS results, it would be challenging to provide a promising novel agent for clinical practice within a short time frame. Several clinical trials for patients with breast cancer [19,20,21,22] have used the degree of pathologic response, such as major pathologic response (MPR) and pathological complete response (pCR), as primary endpoints. Regarding lung cancers, retrospective studies have shown that significant prognostic improvement is observed in patients who showed MPR after neoadjuvant cytotoxic chemotherapy, where MPR is defined as ≤ 10% of the viable residual tumour [23,24]. Although various methods of assessing MPR have been used, they have not been defined in detail [23,24,25,26]. For example, different MPR cut-off values were proposed based on the histological subtypes [27]. In 2020, the International Association for the Study of Lung Cancer published a recommendation for the pathologic assessment of resected specimens after neoadjuvant therapy [28]: a standardised approach is recommended to assess the percentages of (1) viable tumour, (2) necrosis, and (3) stroma (including inflammation and fibrosis) with a total adding up to 100%, and the definition of MPR is ≤10% of viable tumour in the primary tumour bed. Since the pathologic response of resected specimens can be assessed only after the completion of surgery, it is essential to establish predictive markers before therapeutic administration in the selection of patients who are expected to benefit and not be harmed from perioperative treatment.

## 5. Clinical Trials of Neoadjuvant Mono- or Dual ICI Therapy

Various clinical trials of neoadjuvant ICI monotherapy or dual ICI therapy are being conducted. Among these trials, the results of five trials have been reported: four ICI monotherapy trials and one dual ICI therapy trial (Table 2). The proportion of patients who could not undergo surgery ranged from 0 to 12% in the four ICI monotherapy trials but was 19% in the dual ICI therapy trial. Complete resection (R0 resection) was achieved in more than 90% of the patients. The MPR rate ranged from 21% to 45% in all trials in which more than two doses of ICI were scheduled. However, none of the patients showed an MPR in the PRINCEPS trial, in which only one dose of atezolizumab was administered [29].

The NEOSTAR trial is a randomised phase 2 trial of nivolumab monotherapy (the nivolumab arm) or nivolumab plus ipilimumab (the nivolumab plus ipilimumab arm) followed by surgery in patients with clinical stage I to IIIA NSCLCs [30]. The incidence of TRAEs of ≥G3 was equivalent between the two arms (13% in the nivolumab arm vs. 10% in the nivolumab plus ipilimumab arm). However, MPR and pCR were more prevalent in the nivolumab plus ipilimumab arm than in the nivolumab arm: 38% vs. 22% and 29% vs. 9%, respectively (not statistically significant).

**Table 2 cancers-13-04035-t002:** Results of clinical trials using neoadjuvant ICI-mono or -dual therapy.

Registration #	Trial & Stage	Neoadjuvant Therapy	N (Plan)	N (Reported)	Delay of Surgery (%)	Failure to Surgery (%)	R0 Resection (%)	TRAE (≥G3) (%)	MPR (%)	pCR (%)	Survival	Status	Ref
NCT02259621	Johns Hopkins Univ. (p2) IB (>4 cm) to IIIA	nivolumab (twice)	30	22	0	0	95	Preope: 4.5	45	15	Median RFS: NR 18 m RFS: 73%	On going	[31]
NCT02927301	LCMC3 (p2) IB to IIIA, IIIB (T3N2, T4 (size))	atezolizumab (twice)	180	181	12	12	92	Preope: 6 Postope: 14	21	7	1 y DFS: 85% 1 y OS: 95%	On going	[32]
NCT02994576	PRINCEPS (p2) IA (≥2 cm) to IIIA(non-N2)	atezolizumab (once)	60	30	0	0	97	0	0	No data	No data	On going	[29]
NCT03030131	IONESCO (p2) IB to IIIA	durvalumab (3 times)	81	46	No data	0	90	ICI-related: 0 (Death:9)	No data	No data	Median OS/DFS: NR/NR 18 m OS/DFS: 89%/70%	Terminated (mortality *)	[33]
NCT03158129	NEOSTAR (p2) I to IIIA	nivolumab (3 times) or nivolumab (3 times) + ipilimumab	44	44	22	Nivo: 4 N + I: 19	100	Nivo: 13 N + I: 10	Nivo: 22 N + I: 38	Nivo: 9 N + I: 29	Median OS/RFS: NR/NR	On going	[30]

#, number; p2, phase 2; Nivo, nivolumab; R0 resection, complete resection; TRAE, treatment-related adverse event; MPR, major pathologic response; pCR, pathological complete response; RFS, recurrence-free survival; OS, overall survival; DFS, disease-free survival; NR, not reached; N + I, nivolumab + ipilimumab; *, an excess in 90-day postoperative mortality (4 deaths, 9%).

## 6. Clinical Trials for ICI Combination Therapy with Chemotherapy or Chemoradiotherapy

Tumours lacking an immune response are known as ‘cold tumours’. Conventional therapies, such as chemotherapy and radiation therapy, are known to turn ‘cold tumours’ into ‘hot tumours’ with immune responses elicited by the tumour antigens released from cancer cell deaths (i.e., immunogenic cell death), thus increasing the therapeutic effects of ICI [34].

Various neoadjuvant ICI combination therapies have been widely proposed (Appendix A). Phase 3 trials are underway that are exploring combination therapies with ICI against conventional chemotherapy (Appendix A): Checkmate 816 (NCT02998528: nivolumab plus PT-DC), KEYNOTE-671 (NCT03425643: pembrolizumab plus PT-DC with adjuvant pembrolizumab), IMpower030 (NCT03456063: atezolizumab plus PT-DC), AEGEAN (NCT03800134: durvalumab plus PT-DC), and Checkmate 77T (NCT04025879: nivolumab plus PT-DC). Among them, the recent results of the phase 3 trial of Checkmate 816 showed that the proportions of failure to undergo surgery, R0 resection, and TRAE of ≥G3 were equivalent between the combination therapy of nivolumab plus PT-DC and PT-DC alone (16% vs. 21%, 83% vs. 78%, and 19% and 21%, respectively). Nevertheless, the MPR and pCR rates were significantly higher in the combination therapy with nivolumab plus PT-DC than in the PT-DC therapy alone: 36.9% vs. 8.9% (*p* < 0.0001) and 24% vs. 2.2% (*p* < 0.0001), respectively [35] (Table 3). Survival data of this phase 3 trial are not currently available.

A phase 2 NCT03480230 trial exploring the combination therapy of PT-DC with avelumab was terminated because of the low response rate (Table 3). In the other phase 2 trials, the proportion of patients who could not successfully undergo surgery ranged from 3% to 27%, and R0 resection was achieved in 87–100% of the patients. An MPR could be achieved in 57–83% of the patients, and a TRAE of ≥G3 was observed in >27% of patients. The results of the NADIM trial exploring the neoadjuvant combination therapy of carboplatin/paclitaxel plus nivolumab before surgical resection followed by adjuvant nivolumab showed favourable PFS rates of 95.7% and 77.1% at 1 and 2 years, respectively [36].

The ‘abscopal effect’, which is the effect of ionising radiation ‘at a distance from the irradiated volume but within the same organism’, was first reported in 1953 [37]. This phenomenon was revealed to be immune-mediated [38] and has been observed in combination therapy trials when ICIs are administered sequentially or concurrently with radiotherapy [39]. The PACIFIC trial examined the benefits of durvalumab maintenance therapy after concurrent CRT [8]. The results showed that durvalumab maintenance therapy significantly prolonged both PFS (median PFS, 17.2 vs. 5.6 months, HR = 0.51, with 95% CI: 0.41–0.63) [7] and OS (median OS, not reached vs. 29.1 months, HR = 0.69, 95% CI: 0.55–0.86) [40], indicating the usefulness of sequential ICI therapy after CRT.

Several multimodal approaches, in which radiotherapy is added to a combination therapy of ICI and conventional chemotherapy, have been used to improve treatment effects (Appendix A) further. Interim analysis of a phase 2 trial exploring a multimodality therapy using durvalumab, PT-DC, and radiotherapy (45 Gy) (NCT03694236) indicated high MPR and pCR rates of 72.7% and 36.4%, respectively, along with a relatively low rate of TRAEs of ≥G3 (7%) [41] (Table 3). We are also conducting a multicentre, prospective, single-arm, phase 2 trial of neoadjuvant concurrent chemo-immuno-radiation therapy (carboplatin plus paclitaxel and durvalumab with 50 Gy radiation therapy) followed by surgical resection and adjuvant immunotherapy for resectable stage IIIA-B (discrete N2) NSCLC (WJOG12119L: SQUAT trial) (Japic-CTI-195069) [42] (Appendix A).

**Table 3 cancers-13-04035-t003:** Results of clinical trials of neoadjuvant therapy of combination regimens of ICIs.

Registration #	Trial & Stage	Neoadjuvant Therapy	N (Plan)	N (Reported)	Delay of Surgery (%)	Failure to Surgery (%)	R0 Resection (%)	TRAE (≥G3) (%)	MPR (%)	pCR (%)	Survival	Status	Ref
NCT02998528	Checkmate 816 (p3) IB to IIIA	Nivolumab + PT-DC vs. PT-DC	358	358	21 vs. 18	16 vs. 21	83 vs. 78	G3–4: 19 vs. 21	36.9 vs. 8.9 (*p* < 0.0001)	24 vs. 2.2 (*p* < 0.0001)	No data	On going	[35]
NCT02716038	Columbia Univ. (p2) IB to IIIA	atezolizumab + CBDCA/Nab-PTX	30	30	0	3	87	≥50	57	33	Median OS/DFS: NR/17.9 m	On going	[43]
NCT02572843	SAKK16/14 (p2) IIIA (pN2)	durvalumab + CDDP/DTX	68	68	No data	19	No data	Any: 88.1	60	18.2	Median OS/EFS: NR/NR 1 y EFS: 73.3%	On going	[44]
NCT03081689	NADIM (p2) IIIA (pN2)	nivolumab + CBDCA/PTX	46	46	0	11	100	30	83	63	Median PFS/OS: NR/NR 1 y PFS:95.7% 2 y PFS:77.1%	On going	[36]
NCT03480230	American Univ. of Beirut Medical Center (p2) II or IIIA	avelumab + PT-DC	60	15	No data	27	No data	27	No data	9	Median OS/RFS: NR/NR	Terminated (lower response *)	[45]
NCT03694236	Yonsei Univ. (p2) III (N2)	durvalumab + CBDCA/PTX + RT 45 Gy	39	14	No data	8	100	7	72.7	36.4	No data	On going	[41]

#, number; R0 resection, complete resection; TRAE, treatment related adverse event; MPR, major pathologic response; pCR, pathological complete response; p3, phase 3; p2, phase 2; PT-DC, platinum-based doublet chemotherapy; CBDCA, carboplatin; PTX, paclitaxel; CDDP, cisplatin; DTX, docetaxel; RFS, recurrence-free survival; OS, overall survival; DFS, disease-free survival; NR, not reached; *, there was one radiological complete response and three partial responses that were observed among the first 15 enrolled patients, but a minimum of six responses are needed to continue to phase 2 of the study.

The results of the clinical trials for ICI mono- and dual therapies (Table 2) and ICI combination therapy (Table 3) suggest that the proportions of surgery delay, surgery failure, and complete resection were equivalent between ICI mono-/dual therapies and ICI combination therapy. In addition, although TRAEs of ≥G3 were more frequent in ICI combination therapy (19% to ≥50%) than in ICI mono-/dual therapies (0–14%), the MPR and pCR rates may increase in ICI combination therapy (37 to 83% and 9 to 63%, respectively) compared to those in ICI monotherapy (0 to 45% and 7 to 15%, respectively).

## 7. Identification and Monitoring of MRD

An attempt has been made to verify the progression and prognosis of cancer by quantifying tumour cells and tumour-derived DNA released into the blood (liquid biopsy) [46]. The quantification of the circulating tumour DNA (ctDNA) is expected to be an accurate determinant of the indications for perioperative treatment [47,48]. Since blood sample collection is relatively easy, repetitive assessment is acceptable for detecting disease progression and therapeutic effects. Of note, the Checkmate 816 trial showed that ctDNA clearance was more frequent in patients who received neoadjuvant nivolumab plus PT-DC (56%) than in the neoadjuvant PT-DC alone group (34%). Additionally, patients with ctDNA clearance showed higher pCR rates than patients without ctDNA clearance in both treatment groups: 46% vs. 0% in the nivolumab plus PT-DC group, respectively, and 13% vs. 3% in the PT-DC alone group, respectively [35].

The MeRmaiD-2 trial enrolled patients with stage II–III NSCLC who had completed curative-intent therapy (complete resection plus optional neoadjuvant and/or adjuvant therapy) during a 96-week surveillance period (Table 1 and Figure 2, NCT04642469) [49]. During this surveillance period, patients were monitored regularly for MRD emergence via ctDNA analysis using personalised MRD panels. Eligible patients for whom the presence of MRD was confirmed were randomised 1:1 to receive durvalumab or placebo.

## 8. New ICI Agents

Several new ICI agents are being examined in phase 2 trials as neoadjuvant ICI combination therapy (Table 4). Relatlimab is a monoclonal antibody for lymphocyte activation gene 3 (Lag-3), which negatively regulates T lymphocytes by binding to the extracellular domain of the ligand [50]. Oleclumab is an antibody against 5’-nucleotidase ecto, also known as CD73, which binds to CD73 and inhibits the production of immunosuppressive adenosine [51]. Monalizumab is an inhibitor of CD94/NK group 2 member A (NKG2A), an immune checkpoint molecule expressed on tumour-infiltrating cytotoxic T cells and natural killer cells [52]. Tiragolumab is a new immune checkpoint inhibitor blocking the interaction between T-cell immunoreceptors with immunoglobulin and immunoreceptor tyrosine-based inhibitory motif domains (TIGIT) and CD155 (RVR) [53]. Canakinumab is a monoclonal antibody that neutralises IL-1β activity by blocking its interaction with the IL-1 receptor expressed on activated cytotoxic T cells and Tregs [54]. Although canakinumab is a cancer immunotherapy drug (immunomodulator) but not an ICI, a phase 2 trial, Canopy-N, is under way to explore a neoadjuvant monotherapy of canakinumab and combination therapy with pembrolizumab in patients with stage IB to IIIA NSCLC (non-N2 or T4) [55].

## 9. Conclusions and Future Perspectives

Several phase 3 trials using ICIs in the preoperative, postoperative, and both settings are being conducted. The primary results are promising regarding the efficacy of the introduction of ICI in the perioperative phase. The final results of these trials may have a significant impact on the treatment strategies for patients with resectable NSCLC. In addition, the usefulness of ctDNA-based monitoring for MRD should also be substantiated by phase 3 trials to identify patients who genuinely need perioperative therapy with ICI. This approach may provide better clinical outcomes by intensifying the treatment for patients with a high probability of relapse and who are avoiding the unnecessary administration of additional adjuvant chemotherapy (Figure 3).

However, there are several concerns associated with ICI-containing therapies, including the optimisation of ICI administration methods (dosage, dose interval, dose frequency), the optimisation of combination therapies (appropriate regimen and administration methods), and the improved management of ICI-related side effects. These issues should be addressed in basic research and clinical trials. Aggregation of these results would significantly enhance the clinical outcomes of patients with resectable NSCLC.

In conclusion, the initial results of clinical trials on perioperative therapies using ICIs in patients with lung cancer show improved clinical outcomes compared to the current standard of care. Several early phase trials are also investigating the efficacy of novel ICIs. The development of appropriate patient selection methods for perioperative ICIs, such as MRD detection via ctDNA assay, is warranted to maximise the therapeutic effect of perioperative ICIs and to avoid unnecessary administration.

## Figures and Tables

**Figure 1 cancers-13-04035-f001:**
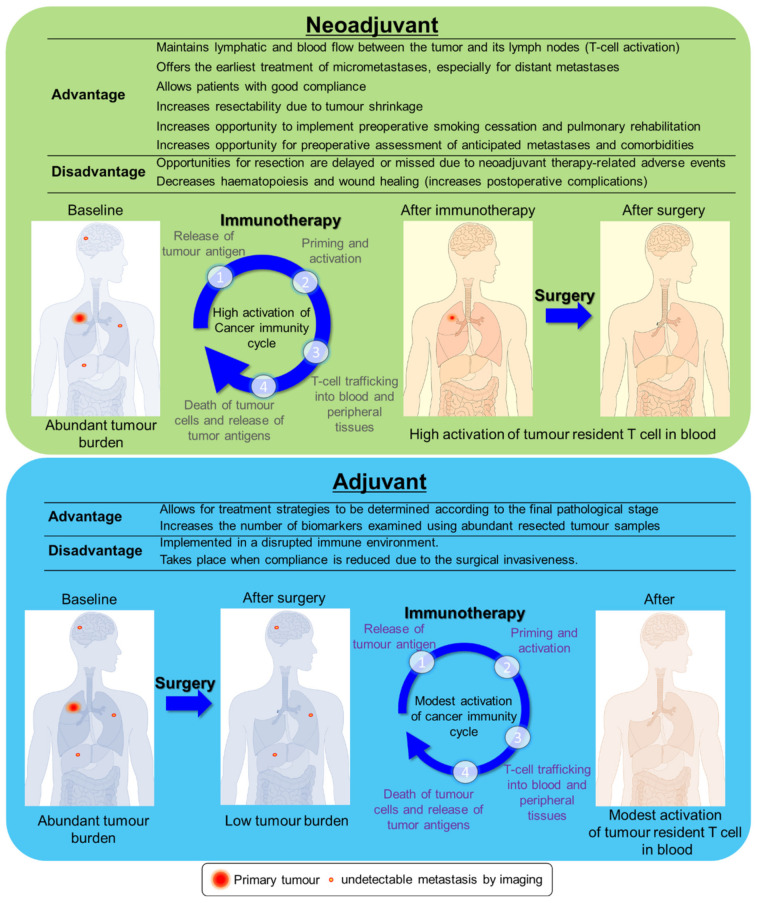
Comparison of neoadjuvant and adjuvant immunotherapies. Neoadjuvant immunotherapy is performed under an abundant tumour burden, which may promote high activation of cancer immunity (upper panel). In contrast, adjuvant immunotherapy is performed under a low tumour burden, but immunotherapy may induce enough efficacy to only control residual disease (lower panel).

**Figure 2 cancers-13-04035-f002:**
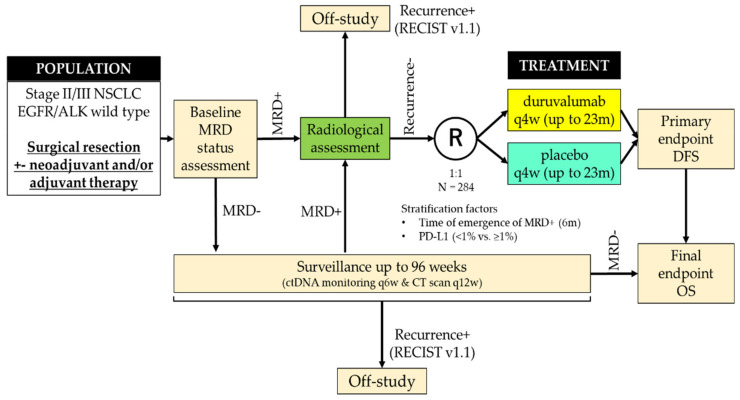
Study design of the MeRmaiD-2 trial (modified from the presentation of Spigel et al. [49]). The MeRmaiD-2 trial (NCT04642469) enrolled patients with stage II–III NSCLC after complete resection plus optional neoadjuvant and/or adjuvant therapy. Eligible patients in whom the presence of MRD was confirmed by regular monitoring for minimal residual disease (MRD) emergence via circulating tumor DNA (ctDNA) analysis during the surveillance period were randomised to receive durvalumab or placebo.

**Figure 3 cancers-13-04035-f003:**
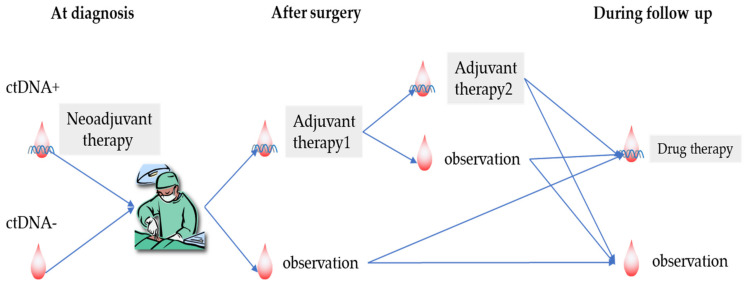
ctDNA-guided perioperative management in the future. This figure demonstrates a future perioperative treatment strategy for patients with resectable NSCLC according to the minimal residual disease assessment. The treatment strategy for neoadjuvant and adjuvant therapies will be determined based on the status of circulating tumour DNA (ctDNA) at diagnosis, after surgery, and during the surveillance period.

**Table 1 cancers-13-04035-t001:** Phase 3 clinical trials of adjuvant therapy using ICIs.

Registration #	Trial	Therapy	N	Pretreatment	Experimental Arm	Control Arm	Primary Endpoint	Stage	Country
NCT02273375	BR.31	ICI mono	1360	Yes/No PT-DC	durvalumab 1 year	Placebo	DFS	pIB to IIIA	Global
NCT02486718	IMpower010	ICI mono	1280	Yes/No PT-DC	Atezolizumab 1 year	BSC	DFS	pIB to IIIA	Global
NCT02504372	PEARLS/KEYNOTE-091	ICI mono	1177	Yes/No PT-DC t	pembrolizumab 1 year	Placebo	DFS	pIB to IIIA	Global
NCT02595944	ANVIL	ICI mono	714	Yes/No PT-DC	nivolumab 1 year	Observation	DFS/OS	pIB to IIIA	US
NCT04642469	MeRmaiD-2	ICI mono	284	Yes/No PT-DC	durvalumab	Placebo	DFS in PD-L1 TC ≥ 1%	II to III without positive EGFR/ALK	Global
NCT04385368	MeRmaiD-1	ICI chemo	322	No	durvalumab + standard of care chemotherapy	Placebo + standard of care chemotherapy	DFS	II to III without positive EGFR/ALK	Global
NCT04564157	NADIM-ADJUVANT	ICI chemo	210	No	Nivolumab + CBDCA/PTX (4 times) Maintenance: nivolumab (6 times)	Nivolumab + CBDCA/PTX (4 times) Maintenance: Observation	DFS	pIB (≥4 cm) to IIIA	Spain

#, number; ICI, immune checkpoint inhibitor; PT-DC, platinum-based doublet chemotherapy; Sq, squamous cell carcinoma, CDDP, cisplatin; CBDCA, carboplatin; PTX, paclitaxel; BSC, best supportive care; DFS, disease-free survival; OS, overall survival; TC, tumor cells.

**Table 4 cancers-13-04035-t004:** Clinical neoadjuvant trials using new ICI agents.

Registration #	Trial	Therapy	Phase	N	Stage	New Agents (Target)	Experimental Arm	Primary Endpoint	Country
NCT04205552	NEOpredict-Lung	ICI dual	2	60	IB to selected IIIA	relatlimab (Lag-3)	Arm A: nivolumab(twice) Arm B: nivolumab + relatlimab (twice)	Feasibility	Belgium Germany Netherlands
NCT03794544	NeoCOAST	ICI dual	2	80	I (>2 cm) to IIIA (single N2 ≤ 3 cm)	oleclumab (CD73) monalizumab (NKG2A)	Arm A: durvalumab Arm B: durvalumab + oleclumab Arm C: durvalumab + monalizumab Arm D: durvalumab + danvatirsen	MPR	Global (Western Countries)
NCT04832854	GO42501	ICI dual + Chemo	2	82	II to IIIB(T3N2)	tiragolumab (TIGIT/RVR)	PD-L1 high: Atezolizumab + tiragolumab (4 times) PD-L1 All comers: Atezolizumab + tiragolumab + PT-DC (4 times)	1. Surgical delays, 2. Complications, 3. Cancellations of surgery, 4. AE, 5. MPR	US Spain Switzerland
NCT03968419	CANOPY-N	ICI + IM	2	110	IB to IIIA (non-N2 nor T4)	canakinumab (IL-1β)	Arm A: canakinumab Arm B: canakinumab + pembrolizumab Arm C: pembrolizumab	MPR	Global

#, number; ICI, immune checkpoint inhibitor; IM, immunomodulator; MPR, major pathologic response; AE, adverse event; danvatisen, a signal transducer and activator of transcription 3 (STAT3) transcription factor inhibitor.

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
