# Peer review of "Perioperative Therapy for Non-Small Cell Lung Cancer with Immune Checkpoint Inhibitors"

_cancers, 2021, doi:10.3390/cancers13164035_

Round 1
Reviewer 1 Report
The manuscript summarized the ongoing trials for adjuvant and neoadjuvant immune checkpoint inhibitors therapies on non-small cell lung cancer. It may give some ideas for clinical therapy.
Can authors add new ICIs target antigen in the table? Are these new agents using in perioperative therapy?
It is better if the authors can also put ICIs in neoadjuvant for ongoing Phase 3 clinical trial in table 3.
In part 5, surrogate or predictive pathological markers of therapeutic effect of ICI, even though the authors did not identify the surrogate or predictive pathological biomarkers after ICIs treatment. Thus, it is not relevant to the main content, better to remove this paragraph.
Author Response
Please find our response to the valuable comments of reviewers 1 and 2 in the attached Word file.

Reviewer 2 Report
In this work, Sohet al. reviewed the perioperative treatment using ICIs in patients with NSCLC. The authors did a good job of reviewing relevant publications, organizing important data, discussing future perspectives. The paper is well written with the help of Editage. Here are some minor comments.
- The captions of Fig. 2-3 are too simple to thoroughly understand the figure on its own.
- Many sections are lacking, such as Author Contributions, Funding, Conflicts of Interest, etc.
- 1 background is kind of bright, so it is not easy to read the text. Therefore, it is suggested to enlarge the font size and change the background.
- Table 1 and 4, is there any reference to support, similar to other tables?
Author Response

(The authors gave the same response as above.)
